# Cognitive Fusion and Emotional Eating among Adolescents with Obesity: A Preliminary Cross-Sectional Study

**DOI:** 10.3390/ijerph192214855

**Published:** 2022-11-11

**Authors:** Anna Guerrini Usubini, Michela Bottacchi, Diana Caroli, Gianluca Castelnuovo, Alessandro Sartorio

**Affiliations:** 1Department of Psychology, Catholic University of Milan, 20123 Milan, Italy; 2Psychology Research Laboratory, Istituto Auxologico Italiano, Istituto di Ricovero e Cura a Carattere Scientifico (IRCCS), 20145 Milan, Italy; 3Istituto Auxologico Italiano, Istituto di Ricovero e Cura a Carattere Scientifico (IRCCS), Experimental Laboratory for Auxo-Endocrinological Research, 28824 Piancavallo-Verbania, Italy; 4Istituto Auxologico Italiano, Istituto di Ricovero e Cura a Carattere Scientifico (IRCCS), Experimental Laboratory for Auxo-Endocrinological Research, 20145 Milan, Italy

**Keywords:** childhood obesity, adolescents, cognitive fusion, experiential avoidance, emotional eating, Acceptance and Commitment Therapy

## Abstract

Cognitive fusion and avoidance are supposed to exert a key role in the development and maintenance of disordered eating behaviors related to obesity, such as emotional eating. A large portion of the research has focused on adult populations, while few data are available on adolescents so far. The current cross-sectional study is intended to explore the association between cognitive fusion, avoidance, and emotional eating in a sample of fifty-six Italian adolescents (13–17 years) with obesity (body mass index > 97th centile). For this purpose, participants attending a 3-week body weight reduction program were assessed using demographical, physical, and clinical data. A multivariate linear regression model was performed in order to preliminarily investigate the predictive role of cognitive fusion on emotional eating, controlling for possible confounding factors. Results showed a significant association between cognitive fusion and emotional eating. Regression revealed that cognitive fusion was a significant contributor for explaining emotional eating (controlling for sex) [R^2^ = 0.551; Adjusted R^2^ = 0.534; F(2,53) = 32.5; *p* < 0.001]. Even if preliminary, our findings suggest a predictive role of cognitive fusion on emotional eating, and also suggest that cognitive fusion can be considered a key component in understanding and addressing of disordered eating behaviors related to obesity. Future replications are required to expand the sample and collect longitudinal data. Intervention programs for childhood obesity could benefit from this line of research.

## 1. Introduction

Cognitive fusion refers to the tendency of an individual to remain trapped in his or her thoughts, and to consider them as if they are literally and objectively true instead of transitory and subjective mental events [1]. Cognitive fusion has been described in Acceptance and Commitment Therapy (ACT) [2,3], one of the “third wave” cognitive behavioral therapies (CBTs). According to the ACT model, fusion refers to the relationship a person has with his or her own cognitive events. “Fused” persons are those who identify themselves with their own thoughts and are unable to consider them as part of their inner experience, rather than descriptions of facts [4]). 

Cognitive fusion is related to experiential avoidance, which includes the unwillingness to remain in contact with aversive internal states and the tendency to take steps to alter the form or frequency of these states, as well as the contexts in which they occur. 

While avoiding internal experiences, people may engage in dysfunctional coping strategies to avoid situations. These include engaging in distracting activities; alternative behaviors with similar functions, such as drug use or alcohol abuse; or disordered eating behaviors, such as emotional eating. 

Emotional eating is defined as the tendency to eat in response to a range of emotional states, including anxiety, depression, anger, and loneliness [5], instead of physiological cues of hunger [6]. Emotional eating is a common phenomenon, both in clinical and non-clinical populations. This became particularly true after the COVID-19 pandemic, which dramatically changed eating behaviors of Italian people, with an increase in emotional overeating [7].

Although emotional eating helps individuals shift their attention away from their unpleasant internal states and temporarily alleviates negative mood states through avoidance [8,9], the temporary emotional relief that it offers can reinforce the maintenance of such eating patterns over time [10]. In addition, it may also lead to undesirable long-term consequences. In fact, emotional eating has been associated with consuming unhealthy food, and, consequently, it has been longitudinally related to weight gain and obesity [11], in both adult and pediatric populations. 

Several relevant studies have reported that cognitive fusion and experiential avoidance play a key role in the development and maintenance of psychopathology [12], including emotional eating. 

By contrast, only a few studies have investigated the role of cognitive fusion on emotional eating in adolescents, in whom emotional eating is common [13], and is also frequently associated with obesity and other obesity-related comorbidities [14].

Taking into account the above considerations, the present preliminary cross-sectional study is intended to explore the association between cognitive fusion and avoidance and emotional eating in a sample of Italian adolescents with obesity, by assessing the hypothesis that cognitive fusion would be a significant predictor of emotional eating.

## 2. Materials and Methods

### 2.1. Participants and Procedures

This study is part of a larger project entitled “the ACTyourCHANGE in Teens” (registered protocol on ClinicalTrials.gov ID: NCT04896372) [15], an RCT which aims to evaluate the effects of an ACT-based intervention on psychological conditions of adolescents with obesity. The intervention was part of a multidisciplinary in-hospital weight loss program. Participants were fifty-six Italian adolescents (12 males, 44 females), the range of age was 13–17 years, and the mean body mass index (BMI: Kg/m^2^) [16]) was 38 (SD = 8.68). Participants were recruited at the Division of Auxology, Istituto Auxologico Italiano IRCCS, Piancavallo (VB), a clinical center for obesity rehabilitation. Participants were included if they were Italian, were 12–17 years old, and had a BMI > 97th centile [13]. Participants were excluded in case of any form of physical or mental impairment (according to the DSM-5 criteria). 

Participants were screened for participation in the study after being informed about the research and after obtaining both written informed consent to participate from their parents and assent from the adolescents. A clinical interview was conducted by a psychologist with expertise in clinical psychology. Once enrolled, participants were asked to answer a set of self-report questionnaires at the beginning of a 3-week body weight reduction program. The diagnostic and assessment procedures were performed by researchers and clinical psychologists with specific expertise in clinical settings. 

The study was approved by the Ethical Committee of Istituto Auxologico Italiano, IRCCS, Milan, Italy (approval number: 2021_01_26_03). Research was carried out according to the Declaration of Helsinki and its advancements.

### 2.2. Measures 

Weight and height were measured in order to calculate BMI according to the proper formula: kg/m^2^. Demographical and clinical data were collected via self-report. As far as clinical data are concerned, we administered the following questionnaires.

The Avoidance and Fusion Questionnaire for Youth (AFQ-Y) [14], Italian version [17], was administered in order to assess cognitive fusion. It is a validated and widely used self-report questionnaire that comprises 8 items rated on a 5-point Likert scale, ranging from 0 (not at all true) to 4 (absolutely true). In our sample, Cronbach’s alpha was 0.904.

The Dutch Eating Behavior Questionnaire—emotional eating subscale (DEBQ-EE) [18], Italian version [19], was administered in order to assess emotional eating. It is a validated and widely used self-report questionnaire that comprises 13 items rated on a 5-step Likert scale, ranging from 0 (never) to 4 (almost always). In our sample, Cronbach’s alpha of the total score was 0.966.

### 2.3. Statistical Analysis 

An a priori power analysis was conducted using G*Power 3.1.9.4 for a linear multiple regression: fixed model, R^2^ deviation from zero. Setting a medium-to-large effect size (f^2^ = 0.2), an alpha of 0.05, and a power of 0.80, we found a required sample size of 52. Analyses were conducted using Jamovi (2.3.2).

Descriptive statistics were computed for all demographical, physical, and clinical variables. Pearson correlations were determined in order to assess the relations between all the continuous variables and to identify possible significant covariates to introduce into the model. An independent sample t-test was used to assess significant differences between males and females in the study variables. A multiple hierarchical linear regression was then computed in order to evaluate the impact of cognitive fusion and avoidance on emotional eating. Possible confounding variables (e.g., sex, age) were entered in the first block. The total score of AFQ-Y was set as the independent variable in the second block. The total score of DEBQ-EE was set as the dependent variable. ΔR2 was used to compare the first and the second block regarding their amount of explained variance of the dependent variable. 

## 3. Results

Descriptive (means and standard deviations) statistics of the study variables are reported in Table 1. Pearson’s correlations showed that cognitive fusion was positively associated with emotional eating, as indicated by a significant and positive correlation between the total score of AFQ-Y and the total score of the DEBQ-EE (r = 0.716; *p* < 0.0001). Age (r = 0.002; *p* = 0.989 and BMI (r = 0.005; *p* = 0.970) was not significantly related to emotional eating. An independent sample t-test showed a significant difference between males (means = 0.558; standard deviation = 0.583) and females (mean = 1.92; standard deviation = 1.30) in terms of emotional eating (t = −5.26; *p* < 0.001), suggesting that females reported higher levels of emotional eating than males. In addition, significant differences were also found between females (mean = 31.4; standard deviation = 15) and males (mean = 18.5; standard deviation = 13.4) in cognitive fusion (t = −2.70; *p* = 0.009).

In line with the aim of the study, which was to explore the association between cognitive fusion and avoidance (AFQ-Y) and emotional eating (DEBQ-EE), a multiple linear regression was run. The model was built to detect the effect of cognitive fusion on emotional eating by controlling for sex, the only demographical variable related to emotional eating, which is entered in the first block. Consequently, the total score of AFQ-Y was entered in the second block. 

According to our findings, cognitive fusion and avoidance (AFQ-Y) were found to be significant contributors to emotional eating (DEBQ-EE) in adolescents with obesity. The first model accounted for a significant amount of variance in emotional eating [R^2^ = 0.186; adjusted R^2^ = 0.171; F(1,54) = 12.3; *p* < 0.001]. Then, when the total score of Y-AFQ was entered, the second model explained 55% of the variance for emotional eating [R^2^ = 0.551; adjusted R^2^ = 0.534; F(2,53) = 32.5; *p* < 0.001]. The second model explained 36% more variance than the first [ΔR^2^ = 0.365; F(1,53) = 43.1; *p* < 0.001]. Results are shown in Table 2. 

## 4. Discussion

The aim of the current work was to provide preliminary results of the study about the association between cognitive fusion and avoidance (AFQ-Y) and emotional eating (DEBQ-EE) in a sample of Italian adolescents with obesity. 

The tested model pointed out the initial evidence of the crucial role that cognitive fusion and avoidance play in emotional eating. According to the model, when people are entangled with their thoughts, they are more likely to exhibit emotional eating. Such disordered eating behavior represents an avoidant strategy aiming to turn back unwanted and painful thoughts [20].

Although the relationship between cognitive fusion and disordered eating behaviors had already been reported in adults [21], the current study sought to expand upon previous knowledge regarding the predictive role of cognitive fusion and avoidance on emotional eating in adolescents with obesity. It is a critical clinical population that suffers from a number of adverse effects on health [22] and requires efforts directed at the prevention and treatment of childhood obesity. In this regard, our results are particularly relevant and offer an important contribution to clinical work by suggesting that treating emotional eating may require focusing on cognitive fusion. 

As mentioned before, cognitive fusion is specifically targeted by Acceptance and Commitment Therapy (ACT) [2,3]. ACT is a psychotherapy that emphasizes the relationship that a person has with their thoughts by promoting distancing from thoughts rather than changing their contents. Such a process has been defined as cognitive defusion. Cognitive defusion techniques in therapy attempt to alter the undesirable functions of thoughts and other private events, rather than trying to alter their form or frequency. In other words, ACT attempts to change the way one interacts with or relates to personal thoughts by creating contexts in which their unhelpful functions are diminished. Cognitive defusion allows people to focus on meaningful behaviors and reduce the influence of dysfunctional thoughts. 

ACT has been broadly and successfully applied in a wide range of psychopathologies and psychological difficulties, including eating behaviors. In this regard, promising results were achieved, suggesting that ACT is effective in targeting emotional avoidance, which exerts a core role in the development and maintenance of eating pathologies [23]. In a study addressing the effects of an ACT intervention for eating behaviors and diet quality in adults with obesity, the authors found that ACT was able to reduce emotional eating, increase acceptance of food, and help people to perceive healthy eating as a chosen behavior driven from personal values and goals [23]. Consistent findings were also collected by our research team. In the “ACTyourCHANGE in Teens” study [24], a brief ACT-based psychological intervention, delivered within a three-week body weight reduction program for adolescents with obesity, was capable of producing a significant reduction in emotional eating. Further investigations are required in order to assess the effects of ACT in sustaining weight loss and weight loss maintenance, as well as the effects of reducing emotional eating over time. 

Several limitations of the study should, however, be pointed out. First, it must be kept in mind that the current study is exploratory, and, therefore, interpretation of the results should be conducted cautiously. Second, the cross-sectional nature of the study limits the formulation of causal hypotheses and requires caution in the interpretation of the results. 

Third, the relatively small sample size, with an imbalance concerning sex (forty-four females and twelve males), although this was dependent on the greater percentage of females in our clinical setting, need to be considered in interpreting the results. For example, although we found a sex difference in cognitive fusion and emotional eating which is in line with the literature, we are obliged to exercise caution in interpretation, since the result may be attributable to the higher prevalence of women in the sample instead of a real sex-related difference. In addition, the sample consisted of Italian adolescents recruited from a single clinical center for obesity rehabilitation. This limits the generalizability of the study. Finally, we involved only self-report measures that could be affected by biases, even though we used valid and widely adopted questionnaires.

In order to overcome these limitations, future replications of the study will need to be developed in order to provide longitudinal results on a larger and more gender-balanced study population. 

By contrast, several strengths should be underlined. First, the study population consisted of adolescents with a high rate of obesity (mean BMI = 38). This made the sample of particular interest for the clinical conditions of participants. Second, the recruitment took place in a single clinical center, allowing us to ensure the consistency of the study procedures. Third, the use of objective measures for physical data and well-validated questionnaires for psychological data guaranteed very consistent, precise, and reliable data.

## 5. Conclusions

In conclusion, by addressing the contribution of cognitive fusion and avoidance on emotional eating in adolescents with obesity, our study suggests that cognitive fusion and avoidance should be considered an important target of psychological interventions intended to address disordered eating behaviors related to obesity, although additional confirmations of the predictive role of cognitive fusion on emotional eating via longitudinal studies are needed to confirm these preliminary observations.

## Figures and Tables

**Table 1 ijerph-19-14855-t001:** Descriptive statistics of the sample.

	*n* (%)	Mean ± SD
Sex		
Male	12 (21.4)	
Female	44 (78.6)	
Age (in years)	56	15.7 ± 1.07
Nationality		
Italians	56 (100)	
Weight	56	107 ± 22.8
BMI (Kg/m^2^)	56	38 ± 8.68
Educational level		
High school	56 (100)	
AFQ-Y	56	28.6 (15.5)
DEBQ-EE	56	1.63 (1.31)

Note. AFQ-Y: Avoidance and Fusion Questionnaire for Youth; DEBQ-EE: Dutch Eating Behavior Questionnaire-Emotional Eating.

**Table 2 ijerph-19-14855-t002:** Multiple hierarchical linear regression model examining the independent effect of demographic features and cognitive fusion (AFQ-Y) on emotional eating (DEBQ-EE).

	B	S.E	β	95% CI	*p*-Value
Block 1 Confounding factors					
Sex	0.6596	0.30915	0.505	0.0303–0.980	0.038
Block 2 Cognitive fusion and avoidance					
AFQ-Y	0.0542	0.00826	0.644	0.4470–0.840	<0.001

Note. AFQ-Y: Avoidance and Fusion Questionnaire for Youth.

## Data Availability

Data will be available upon request from author A.G.U. with the permission of author A.S. The data will not be publicly available due to privacy/ethical restrictions.

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
