# Peer review of "Cognitive Fusion and Emotional Eating among Adolescents with Obesity: A Preliminary Cross-Sectional Study"

_ijerph, 2022, doi:10.3390/ijerph192214855_

Round 1
Author Response
Referee 1 (MS: IJERPH-1963832)
Is the article within the scope of the journal? The paper is within the scope of the journal.
The importance of subject matter and purpose? It is an interesting topic using the theory of ACT and the meaning of the concept as cognitive fusion and avoidance to understand address disordered eating behaviour’s related to obesity.
C1. We thank the reviewer for the appreciation of our work and the critical feedback provided that we have used carefully to improve the quality of our manuscript further. Please find our revisions and considerations below.
Abstract
The abstract describes the different parts and the process in an acceptable way.
Adherence, to ethical standards?
The ethical considerations are clearly mentioned.
Contribution of new knowledge?
The article contributes to the knowledge but needs some clarifications.
C2. We thank the reviewer for his / her appreciation of our work
My main concern and query with the article are the following:
Q1. Introduction: The introduction section is well written with a detailed description about ACT. It can be helpful to have one or two paragraphs describing the disordered eating behaviours investigated in the article. And e.g., be more why is obesity and other obesity-related comorbidities included.
A1. Emotional eating (a dysfuctional eating pattern, not an eating disorder) is defined in line 49-51.
Q2. In the paragraphs of the introduction, there could be less abrupt transitions from one paragraph to another. Even within a few paragraphs, the transitions between the content are somewhat abrupt.
A2. Abrupt transitions have been deleted
Q4. Materials and Methods: The method part is a bit unclear about recruiting the participants. However, this is brought up as a good discussion in the discussion section.
A4. As requested, more detailed information concerning the recruitment of the study group have been added in the Materials and Methods section.
Q5. Result: The result mainly consists of two tables. To clarify the result, it could be interesting to add a clarification in text form. One suggestion is to take the first paragraph of the discussion and attach to the result.
A5. As suggested, the first paragraph of the discussion has been reported in the results section.
Q6. Discussion. ACT is not only a theory but also a practice. This practice may not be part of the study's purpose to investigate, but in the discussion, you could have reflected on how the result can contribute to a clearer insight into how the management of cognitive fusion and experiential avoidance could contribute to getting out of one's state of being entangled in one's thoughts and instead becoming more acceptance of oneself and one's experiences. One suggestion is to move the first paragraph to the results section.
A6.See A5.
Q7. References: Recommend reviewing the reference list and some references once more
A7. Thank you for your recommendation. References have been checked carefully.

Reviewer 2 Report
The work presents a very interesting and innovative theme, it is well written and developed. I suggest some adjustments, described below:
In the introduction, include the prevalence of people with these disorders, showing the overview of the object.
In the methods, include the citations of the references used to classify the BMI and other variables
In the methods, bring the details of the questionnaires used: are they validated? Were they created by the team?
In conclusion, bring the research contributions and future perspectives for the study with this theme.
Author Response
Referee 2 (MS: IJERPH-1963832)
The work presents a very interesting and innovative theme, it is well written and developed.
C1. We thank the reviewer for the appreciation of our work and the critical feedback provided that we have used carefully to improve the quality of our manuscript further. Please find our revisions and considerations below.
Q1. In the introduction, include the prevalence of people with these disorders, showing the overview of the object.
A1. As requested, the prevalence of people with these disorders has been added. Please see line 51-54
Q2. In the methods, include the citations of the references used to classify the BMI and other variables
A2. As requested, the requested references have been added. Please see line 128.
Q3. In the methods, bring the details of the questionnaires used: are they validated? Were they created by the team?
A3. All the questionnaires are validated. As you can see in the text, we have reported the references of the original version and the italian validated version of each questionnaire.
Q4. In conclusion, bring the research contributions and future perspectives for the study with this theme
A4. A sentence regarding the research contributions and future perspectives has been added. Please see line 259-261.

Reviewer 3 Report
Summary:
In the current manuscript entitled Cognitive fusion and emotional eating among adolescents with obesity: a cross-sectional study, the investigators present data from the ACTyourCHANGE in Teens; an RCT aimed to evaluate the effects of an ACT-based (Acceptance and Commitment Therapy) intervention on psychological conditions of adolescents with obesity. In this most recent report, the authors seek to explore the association between cognitive fusion and emotional eating in a cross-sectional study with sample of fifty-six Italian adolescents with obesity (mean age 15.7 ± 1.07, 78.6% female, mean BMI 38 ± 8.68), recruited from an in-hospital multidisciplinary weight loss program. With the hypothesis that cognitive fusion would be a significant predictor of emotional eating. Results show that cognitive fusion may be a significant contributor for explaining emotional eating. In this research the investigators offer much needed clinical implications to an area of investigation that has not been thoroughly addressed in the pediatric population. Obesity is a chronic disease. Evaluation and understanding of mental health and eating-related problems in this population is key to improve the effect of obesity care and prevention.
Minor comments are offered below ahead of publication.
Introduction: The introduction is excellent, well written and condensed. To help clarify for the reader you may consider a clarification of the terms used for eating-related problems, consider modifying terms and keep consistency, lines 43, 56, 67 “disordered eating behavior” and lines 62, 64, 67 “eating habits”. I think these terms are meaning the same thing, choose one term and stick to the chosen one throughout the manuscript unless there is a difference you want to highlight. But then that difference needs to be explained.
Materials and Methods: Question: when did the adolescents answer the questionnaires? Had they started in an intervention or was it baseline data, by line 89 I can assume this was baseline data but this could be clarified?
In the discussion lines 193-196 that your research team performed a 3-week ACT-based psychological intervention as a body weight reduction program for adolescents with obesity. Is this the same intervention as in the adolescents in this cross-sectional study? Please explain the intervention in the materials and methods section or refer to a previously published study where this is explained.
Results: In the write-up of results line 131, and 133-134 does M stand for mean and DS for standard deviation? Consider using “Mean ± SEM” or “Mean (SD)”.
Table 2: Is there a graphical way / figure that you could express these numbers?
Discussion: The discussion is excellent but again I think that you could help clarify for the reader by choosing one term for eating-related problems, consider modifying terms and keep consistency.
Firstly, use the term “emotional eating” when referring to the findings on the eating-related problem researched in this study: Lines 258 “eating behaviors”, 159 “disordered eating behaviors”, and 161 “disordered eating patterns” are all related to the findings of this study, and therefore I suggest that the term should be changed to “emotional eating” as that is what was studied in this study.
Overall authors discuss eating-related problems throughout the discussion. A consistency to the term for eating-related problems would clarify to the reader. Please revise and chose one consistent term for line 164 “disordered eating behaviors”, 167 “eating patterns”, 186 “eating behaviors”, 188 and 225 “eating pathologies” if these terms refer to eating-related problems. Also, the abstract has “eating pathology” on line 24.
Limitations are appropriately addressed.
Thank you for this interesting article.
Author Response
Referee 3 (MS: IJERPH-1963832)
In this research the investigators offer much needed clinical implications to an area of investigation that has not been thoroughly addressed in the pediatric population. Obesity is a chronic disease. Evaluation and understanding of mental health and eating-related problems in this population is key to improve the effect of obesity care and prevention.
C1. We thank the reviewer for his / her appreciation of our work and the critical feedback provided that we have used eagerly to improve the quality of our manuscript further. Please find our revisions and considerations below.
Q1. The introduction is excellent, well written and condensed. To help clarify for the reader you may consider a clarification of the terms used for eating-related problems, consider modifying terms and keep consistency, lines 43, 56, 67 “disordered eating behavior” and lines 62, 64, 67 “eating habits”. I think these terms are meaning the same thing, choose one term and stick to the chosen one throughout the manuscript unless there is a difference you want to highlight. But then that difference needs to be explained.
A1. Since the terms actually indicated the same thing, we have uniformed the textby using disordered eating behaviors as general term and emotional eating as specific disordered eating behavior (it is not an eating disorder, technically)
Q2. Materials and Methods: when did the adolescents answer the questionnaires? Had they started in an intervention or was it baseline data, by line 89 I can assume this was baseline data but this could be clarified?
A2. The questionnaire was administered at the beginning of the 3-week body weight reduction program (i.e. baseline).
Q3. In the discussion lines 193-196 that your research team performed a 3-week ACT-based psychological intervention as a body weight reduction program for adolescents with obesity. Is this the same intervention as in the adolescents in this cross-sectional study? Please explain the intervention in the materials and methods section or refer to a previously published study where this is explained.
A3. The mentioned 3-week body weight reduction program in which the ACT-based psychological intervention was carried out have been extensively described in our previous work. Please see:
Guerrini Usubini, A., Cattivelli, R., Bertuzzi, V., Varallo, G., Rossi, A. A., Volpi, C., ... & Sartorio, A. (2021). The ACTyourCHANGE in teens study protocol: an acceptance and commitment therapy-based intervention for adolescents with obesity: a randomized controlled trial. International journal of environmental research and public health, 18(12), 6225.
Q4. Results: In the write-up of results line 131, and 133-134 does M stand for mean and DS for standard deviation? Consider using “Mean ± SEM” or “Mean (SD)”.
A4. As suggested, M and SDhave been described correctly.
Q5. Table 2: Is there a graphical way / figure that you could express these numbers?
A5. We believe that table 2 represents the best solution to express these results.
Q6. Discussion: The discussion is excellent but again I think that you could help clarify for the reader by choosing one term for eating-related problems, consider modifying terms and keep consistency.
Firstly, use the term “emotional eating” when referring to the findings on the eating-related problem researched in this study: Lines 258 “eating behaviors”, 159 “disordered eating behaviors”, and 161 “disordered eating patterns” are all related to the findings of this study, and therefore I suggest that the term should be changed to “emotional eating” as that is what was studied in this study.
Overall authors discuss eating-related problems throughout the discussion. A consistency to the term for eating-related problems would clarify to the reader. Please revise and chose one consistent term for line 164 “disordered eating behaviors”, 167 “eating patterns”, 186 “eating behaviors”, 188 and 225 “eating pathologies” if these terms refer to eating-related problems. Also, the abstract has “eating pathology” on line 24.
A6. See our answer A1.
Q7. Limitations are appropriately addressed.
Thank you for this interesting article.
A7. Thank you for taking the time to careful review our article. Your feedback was very helpful in improving the overall quality of our manuscript.

Reviewer 4 Report
Comments to the authors
Research question: The aim of the study was to investigate the cross-sectional relationship between cognitive fusion and emotional eating among adolescents with obesity. The authors assessed cognitive fusion and emotion eating via self-reported questionnaires (AFQ-Y / DEBQ-EE) in a small sample of Italian adolescents (N = 56). Besides correlational analyses, a linear regression analysis was conducted. Cognitive fusion contributed significantly to emotional eating while controlling for sex. The authors concluded that cognitive fusion may be important to address eating pathology related to obesity and should be targeted in interventions.
Contribution: The authors present an interesting topic. However, I have some major concerns and I am afraid I have to reject the paper. Though, I hope that the authors find the feedback helpful.
First, my main concern is the small sample size. The authors conducted a power analysis, but I am wondering why they set a medium-to-large effect (0.2) and at the same time reported that there are only a few studies on the role of cognitive fusion in adolescents. Why do they set a medium-to-large effect then? Maybe this might be more an exploratory analysis? As the authors only applied self-reported questionnaires and did not conduct a laboratory assessment, it might be more informative to investigate this research question in a large (online) sample (with self-reported data on BMI).
Second, although the authors discussed the imbalance between males and females (12:44) as a limitation of their study, I am afraid that controlling for sex in the analysis is not appropriate / enough. A larger sample size with a balanced sex-distribution might be needed to investigate sex differences.
Third, the authors used a cross-sectional design. However, when predicting an outcome, a longitudinal design is more appropriate.
Fourth, I think that the conclusions of the paper should be drawn more carefully (that cognitive fusion should be targeted in treatment programs (Abstract)) due to the study design.
Author Response
Referee 4 (MS: IJERPH-1963832)
Q1. First, my main concern is the small sample size. The authors conducted a power analysis, but I am wondering why they set a medium-to-large effect (0.2) and at the same time reported that there are only a few studies on the role of cognitive fusion in adolescents.
A1. To the best of our knowledge, our preliminary study was one of the first aimed to study the role of cognitive fusion in severely obese adolescents. This novelty has been appreciated by the other three reviewers evaluating the present manuscript.
Q2. Why do they set a medium-to-large effect then? Maybe this might be more an exploratory analysis? As the authors only applied self-reported questionnaires and did not conduct a laboratory assessment, it might be more informative to investigate this research question in a large (online) sample (with self-reported data on BMI).
A2. Taking under consideration the paucity of research about cognitive fusion, avoidance and emotional eating in adolescents and the small sample size of the available studies, the current study is intended to be an explorative study which provided preliminary results. Future replications are needed to collect additional data and expand the sample.
As far as face-to-face vs online administration of the questionnaires is concerned, our previous experiences with online questionnaires in severely obese children and adolescents have been negative due to the poor reliability of the self-reported data and scarce feed-back, thus determining a pre-selection of the study population. The preliminary value of this study has been underlined both in the title and in the text, andthe need of further confirmation with larger study population has been admitted.
Q3. Although the authors discussed the imbalance between males and females (12:44) as a limitation of their study, I am afraid that controlling for sex in the analysis is not appropriate / enough. A larger sample size with a balanced sex-distribution might be needed to investigate sex differences.
A3. We have underlined the imbalance between males and females as a limitation of our study even if it follows the gender distribution of our clinical population. We have also reported that future replication with larger sample size is required.
Q4. Third, the authors used a cross-sectional design. However, when predicting an outcome, a longitudinal design is more appropriate.
A4. We agree with this specific point raised by the referee. A longitudinal study will be developed as a next step.
Q5. Fourth, I think that the conclusions of the paper should be drawn more carefully (that cognitive fusion should be targeted in treatment programs (Abstract)) due to the study design.
A5. As suggested, the preliminary nature of the study has been remarked and the conclusions have been drawn more carefully.

Round 2
Reviewer 4 Report
I appreciate the time and effort the authors have taken to revise the paper. I clearly see that the authors acknowledged the limitations of their research by adding them to their article. However, my major doubts are still based on the methodology (e.g., imbalanced gender distribution, cross-sectional study) of the paper and this is the reason why I would reject the paper. I am sure that the authors have made thorough thoughts on developing the research question, conducting the research, analysing the data, and writing the MS. Still, I think that the methodology needs to be improved.
Author Response
Thank you for your comments